# Health-Related Quality of Life among Older Adults with Dementia Living in Vietnamese Nursing Homes

**DOI:** 10.3390/ijerph21020135

**Published:** 2024-01-25

**Authors:** Thanh Xuan Nguyen, Anh Huynh Phuong Nguyen, Huong Thi Thu Nguyen, Thu Thi Hoai Nguyen, Hoa Lan Nguyen, Robert Joel Goldberg, Janani Thillainadesan, Vasi Naganathan, Huyen Thi Thanh Vu, Luc Viet Tran, Anh Trung Nguyen

**Affiliations:** 1Department of Geriatrics, Hanoi Medical University, Hanoi 100000, Vietnam; thuhuonglk@hmu.edu.vn (H.T.T.N.); nththu.bvlk2@gmail.com (T.T.H.N.); vuthanhhuyen11@hmu.edu.vn (H.T.T.V.); trunganhvlk@gmail.com (A.T.N.); 2National Geriatric Hospital, Hanoi 100000, Vietnam; tranvietluc@hmu.edu.vn; 3Institute for Preventive Medicine and Public Health, Hanoi Medical University, Hanoi 100000, Vietnam; pn991897@gmail.com; 4Department of Population and Quantitative Health Sciences, University of Massachusetts Medical School, Worcester, MA 01655, USA; hoa.nguyen@umassmed.edu (H.L.N.); robert.goldberg@umassmed.edu (R.J.G.); 5Department of Geriatric Medicine, Centre for Education and Research on Ageing (CERA), Concord Hospital, Sydney, NSW 2139, Australia; janani.thillainadesan@health.nsw.gov.au (J.T.); vasi.naganathan@sydney.edu.au (V.N.); 6Faculty of Medicine and Health, University of Sydney, Sydney, NSW 2006, Australia; 7Neurology Department, Hanoi Medical University, Hanoi 100000, Vietnam

**Keywords:** Vietnam, dementia, older adult, quality of life, nursing home, related factors

## Abstract

Better understanding of the quality of life among nursing home residents with dementia is important for developing interventions. The objectives of this cross-sectional study were to examine factors associated with poor health-related quality of life in older people with dementia living in nursing homes in Hanoi, Vietnam. In-person interviews were conducted with 140 adults who were 60 years and older with dementia, and information about their quality of life was obtained using the Quality of Life in Alzheimer’s Disease (QOL-AD) scale. The sociodemographic and clinical factors associated with poor health-related quality of life (lowest quartile) were assessed through the results of physical tests, interviews with nursing home staff, and review of medical records. The average age of the study sample was 78.3 years, 65% were women, and their average QOL-AD total score was 27.3 (SD = 4.4). Malnutrition, total dependence in activities of daily living, and urinary incontinence were associated with poor quality of life after controlling for multiple potentially confounding factors. Our findings show that Vietnamese nursing home residents with dementia have a moderate total quality of life score, and interventions based on comprehensive geriatric assessment remain needed to modify risk factors related to poor health-related quality of life.

## 1. Introduction

Dementia is a chronic, progressive, and usually irreversible deterioration in cognition that is associated with considerable morbidity and mortality [1,2]. Globally, the number of people affected by dementia has been increasing during the past several decades, over two-thirds of whom live in low- and middle-income countries, including Vietnam [3]. Since there is currently no cure for dementia, care services are focused on improving or maintaining quality of life in adults with dementia [4]. 

Quality of life (QOL) refers to “a person’s perception of their position in life in the context of the culture and value systems in which they live and in relation to their goals, expectations, standards and concerns” [5]. An individual’s quality of life is personal and subjective and differs between individuals due to their values, life circumstances, and socio-cultural factors. However, measuring quality of life in dementia is challenging because people with more severe cognitive dysfunction have more difficulty with giving their own perceptions. There are several self-report and proxy-report instruments developed for the assessment of QOL in people with dementia, and the most commonly used instrument is the Quality of Life in Alzheimer’s Disease (QOL-AD) scale [6]. 

To know what interventions should be focused on to improve quality of life, the factors associated with poor QOL need to be determined. A large number of studies have investigated factors associated with QOL but there is no consensus on variables related to QOL of people with dementia [4,7]. The presence of geriatric syndromes and co-incidence of geriatric syndromes is common and may result in worsening of QOL among people with dementia [6,8,9,10]. However, assessment of geriatric syndromes is often overlooked in studies among this population. Most previous studies have only revealed that variables such as depressive symptoms, behavioral and psychological disturbance, cognitive functions, and dependence in activities of daily living were most consistently related to low QOL of people with dementia [7,11]. 

While there is an increasing number of older individuals living in nursing homes with dementia [12], little is known about their QOL or factors influencing the health-related QOL of older men and women with dementia living in nursing home settings. In a previous study conducted in eight nursing homes in Vietnam, it was shown that older people in nursing homes in Vietnam had a moderate level quality of life [13]. However, there is a lack of data on QOL in people with dementia living in nursing homes in Vietnam. Better understanding of the quality of life among nursing home residents with dementia is important for developing interventions to improve their quality of life and disease management. The objectives of this cross-sectional study were to measure health-related QOL and identify factors associated with poor health-related QOL among older adults with dementia living in several nursing homes in Hanoi, Vietnam. 

## 2. Materials and Methods

### 2.1. Study Setting

A cross-sectional study was conducted on older adults with dementia living in three nursing home facilities (Orihome, Nhan Ai, Dien Hong Nursing Center) in Hanoi, Vietnam from November 2022 through January 2023. 

In Hanoi, there are presently 20 private nursing homes. The nursing homes included in this study were selected using cluster sampling based on their size: (Small cluster): facility with less than 100 residents; (Medium cluster): facility with 100 to 200 residents; and (Large cluster): facility with more than 200 residents. We then used random sampling to select institutions in each cluster. 

Target population: older adults with dementia living in nursing homes in Hanoi, Vietnam.

Source population: older adults living in private nursing homes that were listed in the Report Market outlook 2021 in Hanoi [14].

Sampling frame: 11 private nursing home systems were licensed to operate at the time of the study in 2022 (Table 1). 

The final study sample consisted of one large cluster (Dien Hong Nursing Center), one medium cluster (Nhan Ai Aged Care Center), and one small cluster nursing home (OriHome Aged Care Center). In the Dien Hong Nursing Center, the number of staff was 136 and the yearly mortality rate in the observation period was 6.8%. In the Nhan Ai Aged Care Center, the number of staff was 36 and the yearly mortality rate in the observation period was 8.8%. In the OriHome Aged Care Center, the number of staff was 15 and the yearly mortality rate in the observation period was 8.8%. All three nursing homes have no relationship with hospitals.

### 2.2. Study Inclusion and Exclusion Criteria 

Long-stay residents (of at least 3 months) at the three participating nursing homes were eligible for study consideration if they were 60 years or older and had a diagnosis of dementia according to DSM 5 criteria [15], as determined by a neuropsychiatrist from the research team at the time of study recruitment. 

Exclusion criteria were if participants had severe loss of hearing or vision, were unable to communicate (according to the interRAI Community Health Assessment) [16] or unable to answer questions about QOL, were unwilling to participate in the study, or if their family did not consent for them to take part in the study.

### 2.3. Data Collection 

The research team consisted of five researchers and two neuropsychiatrists. Prior to recruiting study participants, members of the study team completed a training program on participant screening and study data collection methods. 

After initial screening, participants and their families and nursing home care staff received a complete explanation of the purpose, procedures, and risks of the study. The informed consent of the participants and their families was obtained if they showed interest in participating. 

Data were obtained through four approaches: (1) in-person interviews with study participants; (2) examination of the participants, including physical tests such as the Timed Up & Go (TUG) test and a 30 s chair stand test (30 s CST); (3) interviews with the nursing home care staff; and (4) review of nursing home records. Study participants and nursing home care staff were interviewed separately using a structured questionnaire.

#### 2.3.1. Health-Related Quality of Life 

Health-related QOL, this study’s primary outcome, was assessed using the QOL in Alzheimer’s Disease (QOL-AD) scale through the conduct of in-person interviews [17]. The QOL-AD consists of 13 items (energy, physical health, mood, memory, living situation, family, friends, marriage, self as a whole, ability to do things for fun, ability to do chores, and views of money and life), each of which is rated on a four-point scale: 1—poor, 2—fair, 3—good, and 4—excellent. The total QOL-AD score is the sum of scores from 13 items which range from 13–52 points, with higher scores indicating higher QOL. The internal consistency for the QOL-AD was determined using Cronbach’s alpha. For all 13 items on the scale, the α was 0.77; values above α = 0.70 are considered as evidence of acceptable internal consistency. Each resident’s total QOL-AD score was further categorized according to quartiles as: poor QOL (lowest quartile), moderate QOL (second and third quartiles), and high QOL (highest quartile) based on the distribution of the data and according to what were considered to be clinically meaningful cut-points of QOL.

#### 2.3.2. Other Study Variables

Nursing home paper records were reviewed by members of the research team for the following data on the long-stay residents: age (classified into 3 groups: 60–69 years old, 70–79 years old, 80 years old and over), gender (male, female), marital status (married, divorced/separated, widowed, never married), education level (no formal, primary level, secondary level, college degree or higher), current smoking and drinking, polypharmacy (concurrent use of 5 or more medications [18]), and comorbidity (defined as the co-occurrence of more than one disease). 

Data on each resident’s level of cognition was determined using the Mini-Mental State Examination (MMSE) through in-person interviews by the research team. It is an 11-question measure that tests domains of cognitive function: orientation, registration, attention and calculation, recall, and language. A lower score of MMSE is indicative of greater cognitive decline [19]. Physical function was assessed using the TUG and 30 s CST. For the TUG, participants performed the test twice at the participant’s normal speed and their best time of completion was recorded. [20]. The 30 s CST consists of counting the number of sit-stand-sit cycles completed during the 30 s test [21], where a cutoff score of 11.25 indicates high risk of falls [22].

Nursing home care staff were interviewed to assess the presence of geriatric syndromes in study participants. These included nutritional status, functional ability, incontinence, sleep disturbance, and behavioral and psychological symptoms of dementia. Nutritional status was determined using the Mini Nutritional Assessment Scale-Short Form (MNA-SF), an instrument with scores ranging from 0 to 14 points. A total score of MNA-SF <8, 8–11, and >11 indicates malnutrition, risk of malnutrition, and no malnutrition, respectively. [23]. Functional ability was assessed using the activities of daily living (ADLs) Barthel index, which measures the extent to which somebody can function independently and has mobility in their ADLs with 5 categories (independent: 100 points, slight dependency: 91–99 points, moderate dependency: 61–90 points, severe dependency: 21–60 points and total dependency: 0–20 points) [24]. Presence of incontinence was determined using three Incontinence Questions (3IQ) [25]. Sleep disturbance was assessed using the Pittsburgh Sleep Quality Index (PSQI), which contains 19 questions with a range of 0–21 points, where a score greater than five demonstrates poor sleep quality [26]. Number of geriatric syndromes was recorded, including polypharmacy, poor sleep quality, malnourished, dependence in ADLs, urinary incontinence, comorbidity, high risk of falls, and depression. If a participant had 2 or more geriatric syndromes, he or she was considered to be multimorbidity.

Dementia-related characteristics such as feeding difficulties and behavioral and psychological symptoms were also assessed via interviewing the nursing home care staff. Feeding difficulties were assessed using the Edinburgh Feeding Evaluation in Dementia (EdFED) scale, which is an instrument designed to assess eating and feeding problems in people with late-stage dementia; total scores range from 0 to 20, with 20 being the most serious [27]. The Neuropsychiatric Inventory (NPI) questionnaire was used to assess behavioral and psychological symptoms of dementia (BPSD), including agitation/aggression (total scores range from 0 to 96), depression (total scores range from 0 to 96), and delusional symptoms (total scores range from 0 to 108). Each behavioral symptom was calculated using its total score (frequency × severity), with higher scores indicating greater severity [28].

### 2.4. Data Analysis

Baseline characteristics were compared between the adults with poor and moderate/high QOL. These data were presented as percentages for categorical variables and as means ± standard deviation for continuous variables. The distribution of the total QOL-AD score was examined and was found to be normally distributed (Figure 1). We also calculated internal consistency for the QOL-AD by calculating the Cronbach’s alpha (α = 0.70). 

Poor health-related QOL was defined among the group with the lowest quartile of QOLAD scores. Moderate QOL was defined as inclusion in the second and third quartiles of the QOLAD scores and those with a high QOL were categorized as those in the uppermost quartile of the QOLAD scores.

We examined several factors associated with poor health-related QOL (vs. moderate/high QOL) in older adults with dementia using univariate and multivariable adjusted logistic regression models. These factors included sociodemographic characteristics, polypharmacy, PSQI score, malnutrition, dependence in Barthel ADLs, TUG test, Edinburgh score, comorbidities, the agitation, depression, and delusion scores from the BPSD, 30 s CST test, and the MMSE score. Potential factors associated with the QOL-AD score in the univariate analysis at a threshold of *p*-value less than 0.20 were entered in our multivariable adjusted logistic regression models. Data analyses were performed using STATA version 17 (StataCorp. LP, College Station, TX, USA).

## 3. Results

### 3.1. Study Population Characteristics

A total of 164 participants with dementia were eligible for participation. Of these participants, two (1.22%) were not willing to participate, eight (4.88%) had severe loss of vision or hearing or were unable to communicate, two others (1.22%) were transferred to another nursing home center, and twelve (7.31%) had severe dementia and could not answer questions about their QOL. A total of 140 participants were included for the present analysis. The mean (SD) age of the study sample was 78.3 (+/−8.7) years, females accounted for 65%, and half of the participants (51.43%) were widowed. The mean (SD) number of geriatric syndromes per participants was 4.88 (+/−1.22). The prevalence of no formal education among participants was 10.71%.

### 3.2. Health-Related QOL and Factors Associated with Poor QOL

The distribution of health-related QOL is shown in Figure 1. Study participants had an average QOL-AD total score of 27.3 (SD = 4.4), which ranged from 19 to 38 points. Poor QOL was observed among residents’ ability to do chores, having poor memory, state of physical health, and ability to do things for fun. Family, living situation, and mood status were the domains with the highest reported QOL scores, reported as good to excellent. Detailed scores for each item are shown in Table 2. 

In univariate analyses, being malnourished, total dependence in ADLs, urinary incontinence, feeding difficulties, declines in physical function, and increased number of geriatric syndromes were associated with poor health-related QOL (lowest quartile group). The most common geriatric syndromes in the poor QOL group were being malnourished (78.4%), poor sleep quality (73%), urinary incontinence (56.8%), and total dependence in ADLs (40.5%). The number of geriatric syndromes in the poor QOL group was 5.6 (+/−1.1), higher than those in the high/moderate QOL group (4.6 (+/−1.2)). More than 70% of participants in the poor QOL group had five or more geriatric syndromes (Table 3).

In our multivariable adjusted logistic regression model, poor health-related QOL was associated with malnutrition, total dependence in ADLs, and presence of urinary incontinence (Table 4). 

## 4. Discussion

In this cross-sectional study, we investigated the relationship between geriatric syndromes such as malnutrition, total dependence in ADLs, and urinary incontinence and poor health-related QOL among older adults with dementia residing in nursing homes in Vietnam. 

Vietnam is one of the world’s fastest-aging countries with an increasing number of older individuals with dementia [29]. Policy makers are attempting to build health care models to improve the QOL of these individuals [29]. To the best of our knowledge, this is the first study examining health-related QOL among men and women 60 years and older with dementia living in nursing homes in Vietnam. We found that nursing home residents had, on average, a moderate total QoL-AD score. 

The average QOL-AD scores reported in our study were low compared to previous findings in nursing home residents with dementia living in industrialized countries who used similar study questionnaires [4,30]. For example, a study in eight European countries showed that their average QOL-AD total score was 32.5 (SD = 6.3), ranging from 28.2 (6.2) in Estonia to 35.8 (SD = 5.3) in Sweden [4]. The result from nursing homes in the south of the Netherlands showed that the average QOL-AD total score was 37.3 (SD = 4.7) [30]. These differences can likely be explained by the fact that these earlier studies were conducted in industrialized countries that have different social care systems, healthcare organizations, cultural norms, policy frameworks, family involvement during the care process, and attitudes towards nursing home placement [4]. Notably, in Vietnam, individuals with dementia often stay at home and are only placed in nursing home facilities when their cognitive impairment is severe and their caring needs exceed the capacity of their families. The lower average QOL in this study population may also reflect the fact that residents in other countries enter nursing homes at earlier stages of their disease. 

The QoL-AD scale is a widely used instrument of QoL assessment in dementia. Our study is the first categorizing QoL-AD according to quartiles, as follows: poor QOL (lowest quartile), moderate QOL (second and third quartiles), and high QOL (highest quartile). According to this classification, low QoL-AD was associated with adverse events: being malnourished, total dependence in ADLs, urinary incontinence, feeding difficulties, declines in physical function, and increased number of geriatric syndromes. This study focused on identifying factors associated with poor health-related quality of life among older adults with dementia living in several nursing homes in Hanoi, Vietnam. From these results, we will know which interventions nursing homes should focus on developing to improve their quality of life and disease management.

We observed that a high proportion of participants reported lower QOL in domains related to their physical health, ability to do chores and things for fun, and poor memory. While many studies have used proxies to report QOL among individuals with dementia [31], we asked nursing home residents to self-report their QOL. This approach was taken because individuals with dementia and their caregivers may not perceive QOL in the same way [32], and we sought to identify those aspects of living that are most meaningful to individuals with dementia. Our findings suggest that interventions focusing on physical, social, and cognitive domains should be implemented and evaluated for their effectiveness in enhancing the QOL of individuals with dementia living in nursing homes in Vietnam.

Our results confirmed a high prevalence of multimorbidity among older adults with dementia. All participants were reported having at least two geriatric syndromes. The number of geriatric syndromes in the poor QOL group was higher than those in the high/moderate QOL group. Our research findings show support for evidence from previous studies that indicated the presence and co-incidence of geriatric syndromes were highly prevalent among older adults with dementia [6,8,9,10]. A geriatric assessment is often overlooked in many studies among older adults with dementia due to cognitive impairments that make the assessment more difficult and complicated [33]. Our study revealed that geriatric assessments can be applied to older people with dementia in nursing homes and that the assistance of the caregiver plays an important role in the geriatric assessment process. Previous studies showed that older adults with multimorbidity are at greater risk for worse adverse outcomes [34,35,36,37,38]. They may have functional impairment or a poorer QOL and overall health status [37,38]. Furthermore, they experience greater health services use, such as more frequent inpatient hospitalizations, longer lengths of stay when hospitalized, and increased emergency admissions [36,37]. A geriatric assessment should be identified early to build effective strategies for identifying patients with multiple conditions and initiating prevention and intervention strategies.

Our cross-sectional epidemiological study examined the association between low health-related QOL in older individuals with dementia and modifiable risk factors such as their geriatric syndromes. We found malnutrition, total dependence in activities of daily living, and the presence of urinary incontinence were associated with poor QOL, but a longitudinal study would be necessary to analyze this relationship. These findings underscore the importance of conducting a comprehensive geriatric assessment, which encompasses domains including urinary continence, physical function, and nutrition, since it enables the identification of potential problems and facilitates the implementation of various interventions and approaches to enhance nursing home residents’ overall health-related QOL. 

### Study Strengths and Limitations

To the best of our knowledge, this is the first study to evaluate health-related QOL among older individuals with dementia residing in nursing homes in Vietnam. Our findings have important implications for the type of clinical services required for the care of these older adults. The results of this study should be interpreted with caution, however, due in part to the inherent limitations of cross-sectional studies. This study’s sample size was relatively modest, and participants in this study were recruited from nursing homes, potentially limiting their representativeness to older community dwelling residents. 

## 5. Conclusions

Older individuals with dementia residing in nursing homes in Hanoi, Vietnam generally have a moderate total QoL-AD score. Key challenges include their less-than-optimal physical health, ability to perform chores, poor memory, and need for engagement in enjoyable activities. We recommend interventions targeting these areas to enhance the health-related QOL for this population. Additionally, addressing modifiable risk factors such as malnutrition, activities of daily living, and urinary incontinence through comprehensive geriatric assessment and management is crucial for improving their overall QOL.

## Figures and Tables

**Figure 1 ijerph-21-00135-f001:**
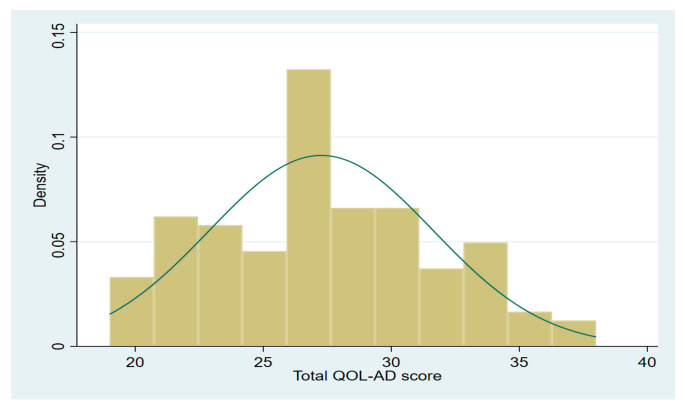
Distribution of total QOLAD scores.

**Table 1 ijerph-21-00135-t001:** Sampling frame of nursing home facilities in Hanoi, Vietnam [14].

N	Facilities	Size
1	Dien Hong Nursing Center (4 campuses) *	L
2	Bach Nien Thien Duc Aged Care Center (3 campuses)	L
3	Lotus Aged Care Center	M
4	Nhan Ai Aged Care Center *	M
5	Tuyet Thai Aged Care Center	M
6	Ha Noi Nursing Center (2 campuses)	M
7	Javilink Nursing Home System (2 campuses)	S
8	KAIGO Nursing Center (2 campuses)	S
9	ALH Nursing Home	S
10	OriHome Aged Care Center *	S
11	FDC Elder Nursing Home	S

Information is referenced from the Report Market outlook for aged care services in Vietnam, through the website and directly calling the facilities. * Selected facilities with updated nursing numbers up to December 2022. S—small, M—medium, L—large.

**Table 2 ijerph-21-00135-t002:** Health-related quality of life among older adults with dementia.

Items	Poor *n* (%)	Fair *n* (%)	Good *n* (%)	Excellent *n* (%)	Mean (SD)
Physical health	36 (25.7)	58 (41.4)	45 (32.1)	1 (0.7)	2.08 (0.78)
Energy	20 (14.3)	72 (51.4)	47 (33.6)	1 (0.7)	2.21 (0.68)
Mood	15 (10.7)	76 (54.3)	47 (33.6)	2 (1.4)	2.26 (0.66)
Living situation	9 (6.4)	78 (55.7)	50 (35.7)	3 (2.1)	2.34 (0.63)
Memory	69 (49.3)	60 (42.9)	10 (7.1)	1 (0.7)	1.59 (0.66)
Family	5 (3.6)	64 (45.7)	64 (45.7)	7 (5.0)	2.52 (0.65)
Marriage	15 (10.7)	74 (52.9)	45 (32.1)	6 (4.3)	2.30 (0.72)
Friends	14 (10.0)	83 (59.3)	39 (27.9)	4 (2.9)	2.24 (0.66)
Self as a whole	8 (5.7)	99 (70.7)	30 (21.4)	3 (2.1)	2.20 (0.57)
Ability to do chores	70 (50.0)	56 (40.0)	13 (9.3)	1 (0.7)	1.61 (0.69)
Ability to do things for fun	33 (23.6)	82 (58.6)	24 (17.1)	1 (0.7)	1.95 (0.66)
Money	25 (17.9)	102 (72.9)	13 (9.3)	0 (0.0)	1.91 (0.52)
Life as a whole	13 (9.3)	105 (75.0)	22 (15.7)	0 (0.0)	2.06 (0.50)
Total score					27.26 (4.37)

**Table 3 ijerph-21-00135-t003:** Characteristics of nursing home residents with dementia according to quartile of health-related quality of life.

Characteristics	Quality of Life		*p*
	Poor (*n* = 37)	Moderate/High (*n* = 103)	
Age, mean, years (SD)	79.3 (8.8)	78.0 (8.6)	0.43 ^a^
Female, *n* (%)	21 (56.8)	70 (68.0)	0.22 ^c^
Education level, *n* (%)			
No formal	4 (10.8)	11 (10.7)	0.98 ^d^
Not finished high school	5 (13.5)	13 (12.6)	
Finished high school	21 (56.8)	51 (49.5)	
College	4 (10.8)	18 (17.4)	
Bachelor’s Degree or Higher	3 (8.1)	10 (9.7)	
Current marital status, *n* (%)			
Married	14 (37.8)	40 (38.8)	0.49 ^d^
Divorced/Separated	1 (2.7)	5 (4.9)	
Widowed	18 (48.6)	54 (52.4)	
Never Married	4 (10.8)	4 (3.9)	
Current smoking, *n* (%)	8 (21.6)	17 (16.5)	0.49 ^c^
Current drinking, *n* (%)	7 (18.9)	14 (13.6)	0.44 ^c^
Polypharmacy, *n* (%)	3 (8.1)	11 (10.7)	0.47 ^d^
Poor sleep quality, *n* (%)	27 (73.0)	71 (68.9)	0.64 ^c^
Malnourished, *n* (%)	29 (78.4)	45 (43.7)	<0.001 ^c^
Total dependence in ADLs, *n* (%)	15 (40.5)	7 (6.8)	<0.001 ^c^
Urinary incontinence, *n* (%)	21 (56.8)	28 (27.2)	0.001 ^c^
Comorbidity, mean (SD)	3.3 (1.7)	3.0 (1.5)	0.47 ^b^
Agitation/aggression, mean (SD)	6.5 (9.6)	7.9 (11.7)	0.68 ^b^
Depression, mean (SD)	5.3 (8.7)	3.1 (4.7)	0.31 ^b^
Delusion, mean (SD)	1.3 (2.0)	0.9 (1.7)	0.07 ^b^
MMSE, mean (SD)	15.1 (4.8)	16.3 (4.1)	0.14 ^a^
TUG, mean (SD)	15.2 (22.4)	16.7 (19.0)	0.36 ^b^
30-s CST, mean (SD)	1.5 (2.5)	4.0 (4.7)	0.007 ^b^
EdFED scale, mean (SD)	7.1 (4.0)	4.9 (4.5)	0.001 ^b^
Number of geriatric syndromes, mean (SD)	5.6 (1.1)	4.6 (1.2)	<0.001 ^a^
Number of geriatric syndromes, *n* (%)			<0.001 ^d^
2	0 (0)	3 (2.9)	
3	0 (0)	14 (13.6)	
4	10 (27)	29 (28.2)	
5	4 (10.8)	35 (34)	
6	15 (40.5)	16 (15.5)	
7	8 (21.6)	6 (5.8)	

^a^ *t*-test; ^b^ Mann–Whitney U test; ^c^ Chi-square test; ^d^ Fisher exact test.

**Table 4 ijerph-21-00135-t004:** Factors associated with poor QOL-AD score among older adults with dementia.

Characteristics	Unadjusted	Multivariable Adjusted
	Odds Ratio (95% CI)	Odds Ratio (95% CI)
Age group (years)		
60–69	1.00	
70–79	0.63 (0.19–2.10)	
≥80	1.30 (0.45–3.74)	
Male	1.62 (0.75–3.49)	
Marital status		
Married	1.00	
Divorced/Separated	0.57 (0.06–5.32)	
Widowed	0.95 (0.42–2.14)	
Never Married	2.86 (0.63–12.98)	
Polypharmacy	0.74 (0.19–2.81)	
Poor sleep quality	1.22 (0.53–2.81)	
Malnourished	4.67 (1.95–11.20)	3.41 * (1.29–9.02)
Total dependence in ADLs	9.35 (3.41–25.67)	4.67 * (1.32–16.47)
Urinary incontinence	3.52 (1.61–7.68)	2.72 * (1.09–6.79)
TUG, second	1.00 (0.98–1.02)	
EDINBURGH, score	1.12 (1.03–1.21)	0.99 (0.88–1.10)
Number of Comorbidity	1.14 (0.90–1.45)	
Agitation/aggression, score	0.99 (0.95–1.02)	
Depression, score	1.06 (1.00–1.12)	1.04 (0.96–1.13)
Delusion, score	1.13 (0.93–1.37)	
30-s CST, time	0.84 (0.75–0.95)	0.94 (0.82–1.07)
MMSE, score	0.94 (0.86–1.02)	1.00 (0.90–1.11)
Number of geriatric syndromes, number	2.05 (1.42–2.96)	

Notes: * *p* < 0.05.

## Data Availability

The datasets generated during and/or analyzed for the current study are available from the corresponding author on reasonable request.

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
