# Peer review of "Health-Related Quality of Life among Older Adults with Dementia Living in Vietnamese Nursing Homes"

_ijerph, 2024, doi:10.3390/ijerph21020135_

Round 1

Reviewer 1 Report

Comments and Suggestions for Authors

The authors are proposing a quality of life evaluation amongst dementia patients living in nursing homes.

The article is well written, providing a good statistical analysis and data interpretation.

However, this manuscript could be improved by adding a paragraph describing the background concerning this topic in Vietnamese nursing homes in the introduction section.

I wish the authors good luck in their future scientific careers.

Author Response

Thank you for this comment. To our knowledge, this is the first study to measure health-related QOL among older adults with dementia living in several nursing homes in Vietnam. We have added 2 sentences in the introduction concerning this topic in Vietnamese nursing homes.

(please see page 2, line 72-75 in track changes).

 « In a previous study conducted in eight nursing homes in Vietnam showed that older people in nursing homes in Vietnam had a moderate level quality of life [14]. However, there is a lack of data on QOL in people with dementia living in nursing home in Vietnam. »

Reviewer 2 Report

Comments and Suggestions for Authors

I consider the article is relevant and may be published in Special Issue “Prevention and Management of Multimorbidity in Older People”, after the minor spelling errors and rephrasing of the section highlighted. 

Regarding the abstract of the article, I think that the authors are trying to emphasize the importance of caring for those who are hospitalized in a center for the elderly. I identify this hypothesis from the way the article is structured.

I also think that the authors paper represents an original contribution, evaluating several care centers that they have catalogued according to size and the number of residents they care for.

As a distinctive element compared to other works, I noticed that the quality of care for patients with Dementia, a chronic pathology, which is frequently neglected, is thoroughly evaluated. A comparison is made between several associated pathologies in patients with Dementia.

My opinion is that the conclusions are relevant for the studied topic and also the manuscript is well structured. The ways in which the quality of life could increase if certain elements that are related to the care of patients with Dementia can be improved are also highlighted.

The sections of the manuscript that in my opinion need rephrasing with proper english (line 60-63), line 248 (attitudes toward needs to be attitude towards), line 295 needs a space between ever,due.

Comments on the Quality of English Language

please find highlighted the sections of the manuscript that in my opinion need rephrasing with proper english (line 60-63), line 248 (attitudes toward needs to be attitude towards), line 295 needs a space between ever,due.

Author Response

Thank you for this comment. We have rephrased with proper english (line 60-63), line 248 (attitudes toward needs to be attitude towards), line 295 needs a space between ever,due as the reviewer suggested.

 (please see page 2, line 75-77 in track changes)

 “Better understanding of the quality of life among nursing home residents with dementia is important for developing interventions to improve their quality of life and disease management”

(please see page 9, line 300 in track changes)

“These differences can likely be explained by the fact that these earlier studies were conducted in industrialized countries that have different social care systems, healthcare organizations, cultural norms, policy frameworks, family involvement during the care process and attitude towards nursing home placement [4]. »

(please see page 10, line 361 in track changes)

“The results of this study should be interpreted with caution, however, due, in part, to the inherent limitations of cross-sectional studies.”

Reviewer 3 Report

Comments and Suggestions for Authors

Dear authors,

Congratulations on your promising results. However, I have a few comments that I would like to discuss with you.

1.     The title indicates a focus on Health-Related Quality of Life, attributed to the utilization of Alzheimer's Disease's QoL assessment. Please explicitly articulate this connection within the manuscript for consistency.

2.     The second and third paragraphs of the introduction seem repetitive and lack clear articulation of study hypotheses or research questions. Revise to eliminate redundancy and explicitly state primary objectives.

3.     If the decision to combine moderate and high groups is not addressed, provide a brief explanation. If analyses showed no differences, consider presenting this in a supplementary section.

4.     Move content from lines 175-178 to the section on geriatric syndrome for improved coherence.

5.     Include informative notes in tables specifying represented data and relevant statistical analyses.

6.     In unadjusted regression analysis, consider including the number of geriatric syndromes

7.     In Table 4, include p-values or bold significant results for emphasis.

8.     Conduct interaction analyses to validate results. Explore how two significant variables interact and maybe analyze three QoL groups based on geriatric syndromes or malnourishment.

9.     Enhance the discussion and conclusion by extending the analysis beyond European scores. It is crucial to avoid generalizing that participants generally have a moderate Quality of Life (QOL), especially when their classification is based on self-identified tertiles. Provide a more nuanced interpretation that considers the self-classification method and explores variations within the tertiles.

Reviewer 4 Report

Comments and Suggestions for Authors

The article deals with a novel and highly complex topic. Introduction The authors use the most widely used instrument for measuring quality of life (QOL). It would be important in this section to differentiate QOL from quality of care. The measurement of results in Nursing homes is a very important topic and QOL can be a dimension, but in general a group of outcomes is used. Study design It would be necessary to have some data on the performance characteristics of the Nursig Home in addition to size, such as staff, mortality, and the relationship with hospitals. Results It is essential to know the patients with dementia who could not participate in the study and the reasons. We recommend classifying patients according to the degree of cognitive impairment. Discussion The relationship between QOL and geriatric syndromes is difficult to interpret in a cross-sectional study, it can be concluded that there is an association between low QOL and geriatric syndromes, a longitudinal study would be necessary to analyze this relationship. In general, national dementia plans aim for governments to give priority to this issue, seeking the maximum adequacy of services and guaranteeing their sustainability. The comparison with European countries is very difficult, in the same study to which the authors refer, there are important differences between the different European countries. Surely the length of stay prior to death of people living in nursing homes can provide information regarding the form of use. Finally, geriatric assessment should be a standard of quality of care in nursing homes.
